# Hexokinase 2 Regulates Ovarian Cancer Cell Migration, Invasion and Stemness via FAK/ERK1/2/MMP9/NANOG/SOX9 Signaling Cascades

**DOI:** 10.3390/cancers11060813

**Published:** 2019-06-12

**Authors:** Michelle K. Y. Siu, Yu-Xin Jiang, Jing-Jing Wang, Thomas H. Y. Leung, Chae Young Han, Benjamin K. Tsang, Annie N. Y. Cheung, Hextan Y. S. Ngan, Karen K. L. Chan

**Affiliations:** 1Department of Obstetrics and Gynecology, University of Hong Kong, Hong Kong, China; mkysiu@gmail.com (M.K.Y.S.); yuxin_jiang2012@126.com (Y.-X.J.); wjj01947@connect.hku.hk (J.-J.W.); thyl@hku.hk (T.H.Y.L.); hysngan@hku.hk (H.Y.S.N.); 2Department of Obstetrics and Gynecology and Cellular and Molecular Medicine, University of Ottawa, Ottawa, ON K1H 8L6, Canada; lemon7time@gmail.com (C.Y.H.); btsang@ohri.c (B.K.T.); 3Chronic Disease Program, Ottawa Hospital Research Institute, Ottawa, ON K1H 8L6, Canada; 4State Key Laboratory of Quality Research in Chinese Medicine, Macau Institute for Applied Research in Medicine and Health, Macau University of Science and Technology, Macao, China; 5Department of Pathology, University of Hong Kong, Hong Kong, China; anycheun@pathology.hku.hk

**Keywords:** HK2, metastasis, stemness, FAK/ERK signaling, ovarian cancer

## Abstract

Metabolic reprogramming is a common phenomenon in cancers. Thus, glycolytic enzymes could be exploited to selectively target cancer cells in cancer therapy. Hexokinase 2 (HK2) converts glucose to glucose-6-phosphate, the first committed step in glucose metabolism. Here, we demonstrated that HK2 was overexpressed in ovarian cancer and displayed significantly higher expression in ascites and metastatic foci. HK2 expression was significantly associated with advanced stage and high-grade cancers, and was an independent prognostic factor. Functionally, knockdown of HK2 in ovarian cancer cell lines and ascites-derived tumor cells hindered lactate production, cell migration and invasion, and cell stemness properties, along with reduced FAK/ERK1/2 activation and metastasis- and stemness-related genes. 2-DG, a glycolysis inhibitor, retarded cell migration and invasion and reduced stemness properties. Inversely, overexpression of HK2 promoted cell migration and invasion through the FAK/ERK1/2/MMP9 pathway, and enhanced stemness properties via the FAK/ERK1/2/NANOG/SOX9 cascade. HK2 abrogation impeded in vivo tumor growth and dissemination. Notably, ovarian cancer-associated fibroblast-derived IL-6 contributed to its up-regulation. In conclusion, HK2, which is regulated by the tumor microenvironment, controls lactate production and contributes to ovarian cancer metastasis and stemness regulation via FAK/ERK1/2 signaling pathway-mediated MMP9/NANOG/SOX9 expression. HK2 could be a potential prognostic marker and therapeutic target for ovarian cancer.

## 1. Introduction

Ovarian cancer is the most lethal of all gynecological malignancies worldwide [1]. Its high mortality is mainly due to the late presentation of symptoms at the advanced stages of the disease, often once the cancer has metastasized [2,3]. Primary treatment mainly involves cytoreductive surgery followed by adjuvant chemotherapy. However, even with optimal treatment, recurrences are common, and the overall prognosis is poor. Continued efforts to identify and develop new target therapies are, therefore, essential.

Unlike most solid tumors, ovarian cancer seldom metastasizes via intravasation. As an intra-abdominal tumor, metastasis arises when exfoliated ovarian cancer cells that have detached from the primary tumor float in the peritoneal fluid and spread throughout the peritoneal cavity [3]. Thus, patients with advanced stages of ovarian cancer are often diagnosed with an accumulation of peritoneal fluid (called ascites), which is correlated with metastasis and chemoresistance. Notably, shedding ovarian cancer cells are often present as single cells and/or multicellular spheroids, and serve as a vehicle for tumor dissemination [3]. Spheroids formed by ascites-derived ovarian cancer cells and ovarian cancer cell lines display upregulated expression of stemness-associated genes and the capacity to form successive generations of spheroids, characteristics that are consistent with the sphere-forming and self-renewing properties of cancer stem-like cells (CSCs) [4]. CSCs are a small sub-population of malignant cells that possess stem-like properties, responsible for cancer initiation and metastasis, and contribute to the ineffectiveness of chemotherapy and cancer recurrence [5,6]. It is thought that some forms of cancer can only be cured by targeting this subset of cells [7]. However, the mechanisms regulating CSCs in ovarian cancer remain unknown.

Reprogramming of glucose metabolism is one of the hallmarks of cancer [8]. In the presence of oxygen, normal cells primarily metabolize glucose to produce pyruvate via glycolysis. Subsequently, most of the pyruvate is oxidized in the mitochondrial tricarboxylic acid cycle through oxidative phosphorylation (OXPHOS) to generate the energy (ATP) needed for cellular processes. In contrast, cancer cells mainly convert pyruvate into lactate with minimal ATP production in the cytoplasm despite normal oxygen conditions, a phenomenon termed aerobic glycolysis, or the Warburg effect [9]. Since the generation of ATP via aerobic glycolysis is minimal in cancer cells, increased glucose uptake and lactate production are both required. Such an altered metabolic phenotype contributes to tumorigenesis and poor cancer prognosis. Therefore, glycolytic enzymes could be exploited to selectively target cancer cells in cancer therapy [9].

Hexokinases (HKs) catalyze the first irreversible enzymatic step in glucose metabolism by phosphorylating glucose to glucose-6-phosphate (G6P) [10]. Five HK isoforms (HKs1–5) have been identified in humans. Overexpression of HK2 has been found in many cancer cells, and forms the basis for ^18^FDG-PET imaging, a common clinical technique used to detect tumors [11]. In addition, HK2 has been shown to promote glycolysis and tumor growth in vivo in glioblastoma [12], medulloblastoma [13] and breast cancer [14]. Notably, HK2 is required for tumor initiation and maintenance in mouse models of K-Ras-driven lung and ErbB2-driven breast cancers [11]. Systemic HK2 deletion has been shown to be therapeutic in mice, without adverse side effects [11].

Since G6P—the product formed by HKs—was higher in metastatic ovarian cancers when compared to normal ovarian tissue and primary ovarian cancers [15], a role for HK2 in ovarian cancer metastasis is suggested. In this study, we focused not only on the clinical significance, but also on the functional roles and downstream mechanisms of HK2 in ovarian cancer. We also assessed the effects of conditioned media derived from ovarian cancer-associated fibroblasts (CAF-CM) on HK2 expression. Our in vitro and in vivo findings demonstrate that HK2 is overexpressed in ovarian cancer, ascites and metastatic foci. This overexpression controls lactate production and promotes metastasis and stemness of ovarian cancer cells through focal adhesion kinase (FAK)/extracellular signal-regulated kinase (ERK1/2) activation-induced matrix metalloproteinase 9 (MMP9)/NANOG/SRY-Box 9(SOX9), as demonstrated in ovarian cancer cell lines and clinical samples. Moreover, we have demonstrated that the induction of HK2 in cancer is partly due to CAF-CM-derived interleukin-6 (IL-6) activation, via its receptor IL-6R. Taken together, this study reveals that HK2 is a potentially useful molecular prognostic marker and a therapeutic target for ovarian cancer.

## 2. Results

### 2.1. Overexpression of HK2 Correlates with Ovarian Cancer Metastasis and Patient Prognosis

By immunohistochemistry, moderate to strong HK2 protein was localized in the cytoplasm in ovarian cancer samples; in contrast, it was barely detectable in benign cystadenomas (Figure 1A). The HK2 immunoreactivity in ovarian cancers was significantly higher than in benign cystadenomas (*p* < 0.001; Appendix A). High HK2 immunoreactivity was significantly associated with a more advanced stage (Stage 4), higher grade (grade 3), and shorter overall and disease-free survival (all *p* < 0.05; Appendix A and Figure 1B). Moreover, statistically higher HK2 immunoreactivity was detected in metastatic foci than their corresponding primary carcinomas (Figure 1C). By multivariate analysis, HK2 expression was a significant independent predictor of disease-free survival (*p* = 0.033; Appendix A). By western blot analysis, we found an up-regulation of HK2 protein expression in ovarian cancer cell lines (OVCAR-3, OVCA429, OVCA433, ES-2, TOV21G, TOV112D, A2780S, and A2780CP), compared to normal ovarian epithelial cell lines (HOSE 6-3 and HOSE 11-12) (Figure 1D).

### 2.2. HK2 Increases Lactate Production

We first detected the specific transient (siHK2; Figure 2A) and stable (shHK2; Figure 2B) knockdown of HK2 in A2780CP and ES-2 cell lines, ovarian cancer cell lines with relatively high HK2 expression. We then examined the effect of HK2 on intracellular lactate production. Results showed that HK2-transiently and stably silenced cells had a significantly reduced lactate level compared to control cells, as assessed by the Lactate Colorimetric Assay Kit II (Figure 2C).

### 2.3. HK2 Augments Cell Migration and Invasion via the FAK/MEK-1/ERK1/2/MMP9 Signaling Pathway

Our finding of statistically higher HK2 immunoreactivity in metastatic foci compared to their corresponding primary carcinomas prompted us to investigate the effect of HK2 on cell migration and invasion. In a wound-healing assay, a slower migration rate was observed in siHK2 ES-2, shHK2 A2780CP, and ES-2 cells, as compared to control cells (Figure 2D). In Transwell migration and invasion assays, significantly reduced migration and invasion were detected in A2780CP and ES-2 cells upon siHK2 (Appendix A) or shHK2 (Figure 2E) knockdown. We next determined whether the glucose analog 2-DG, an inhibitor of glycolysis [16], was able to mimic the effects of HK2 silencing in ovarian cancer cells. A2780CP, ES-2 and OVCA 433 cells were treated with vehicle (water) or low doses of 2-DG (0, 0.5, or 2 mM). Migration and invasion assays revealed significantly reduced migration and invasion in 2-DG treated cells, compared to that of control cells (Figure 2F). An XTT assay revealed no change in cell proliferation 48 h after low doses of 2-DG treatment (Appendix A).

Reduced lactate levels after depletion of HK2 may contribute to its inhibitory effect on cell migration and invasion, owing to the observation that lactate can enhance the motility of head and neck tumor cells [17]. However, investigating the possible molecular pathways by which HK2 mediates its effect on cell migration and invasion is still warranted. FAK is a cytoplasmic protein tyrosine kinase that is overexpressed and activated in several advanced-stage solid cancers, including ovarian cancer [18]. FAK is a key component of cell-matrix adhesion complexes, which together with other signaling and adaptor proteins, such as ERK1/2, act to regulate cancer cell migration and invasion via promoting matrix metalloproteinase (MMP) expression [18]. Binding of urokinase-type plasminogen (uPA), a serine protease, to its receptor uPAR converts proenzyme plasminogen to active plasmin. Plasmin, in turn, remodels the extracellular matrix (ECM) and activates growth factors, results in increased cellular invasion and metastasis [19]. Vascular endothelial growth factor (VEGF) is a pro-angiogenic protein produced by cancer cells and endothelial cells, which plays important roles in ECM degradation and angiogenesis, which in turn affects metastasis, including in ovarian cancer [20]. Interestingly, we found that FAK and ERK1/2 activation was inhibited after knockdown of HK2 in A2780CP and ES-2 cells (Figure 2G). We also found that HK2-silenced A2780CP cells had reduced MMP9, uPA, and VEGF mRNA expression (Figure 2H). A reduction of MMP-9 and uPA protein levels in conditioned media was also detected (Figure 2H). GEPIA further revealed a significant clinical correlation between MMP9, uPA, and VEGF and HK2 expression (*p* < 0.05) in ovarian cancer clinical samples (Figure 2I). In addition, 2-DG inhibited FAK and ERK1/2 activation, and suppressed MMP-9 mRNA expression in OVCA 433 cells (Figure 2J).

To further evaluate the effect of HK2 and its downstream signaling pathways in ovarian cancer cell migration and invasion, we stably transfected SKOV-3 cells, an ovarian cancer cell line with relatively low HK2 expression, with DDK-tagged HK2 plasmid or control empty vector. Ectopic expression of HK2 was detected in HK2-transfected cells [western blot analysis using an anti-DDK antibody (Figure 3A)]. We found that ectopic expression of HK2 enhanced cell migration and invasion (Figure 3B), as well as increased FAK and ERK1/2 activation (Figure 3C), along with increased MMP9, uPA and VEGF mRNA expression (Figure 3D). To unravel the effect of HK2-mediated FAK and ERK1/2 activation on cell migration and invasion, and to elucidate potential links between FAK, ERK1/2, and MMP-9, SKOV-3 cells overexpressed with HK2 were treated with a FAK inhibitor (FAK inhibitor 14), a mitogen-activated protein kinase kinase -1 (MEK-1) inhibitor (U0126), or an anti-MMP9 neutralizing antibody. We found that FAK inhibitor 14 abolished HK2-induced cell migration and invasion (Figure 3B). As determined by western blot analyses and qPCR, FAK inhibitor 14 attenuated HK2-mediated ERK1/2 activation (Figure 3C) and MMP-9 mRNA expression (Figure 3E). Moreover, U0126 abrogated HK2-induced cell migration and invasion, and also inhibited HK2-mediated MMP-9 mRNA expression (Figure 3B,C,E). Inhibition of HK2-induced cell migration and invasion was also observed following treatment with an anti-MMP9 neutralizing antibody (Figure 3F). These findings indicate that FAK activates MEK-1/ERK1/2, which in turn mediates HK2-induced MMP-9 expression. We found that HK2 expression was barely detectable in HOSE 6-3 and HOSE 11-12 cells (Figure 1D), whether induction of HK2 would enhance migration, invasion or other functions, thus progression, of normal ovarian epithelial cells could be further studied in a future study.

### 2.4. HK2 Enhances Cell Proliferation and Anchorage-Independent Growth

We further evaluated the effects of HK2 on ovarian cancer cell growth. Significantly reduced proliferation was observed 11 d after shHK2 knockdown in A2780CP and ES-2 cells, as assessed by cell counting (Appendix A). A soft agar assay revealed that smaller and fewer colonies were formed in A2780CP cells after shHK2 knockdown, whereas and larger and more colonies were formed in SKOV-3 cells with HK2 overexpression (Appendix A), suggesting that HK2 can induce anchorage-independent growth.

### 2.5. HK2 Induces Stemness via the FAK/MEK-1/ERK1/2/NANOG/SOX9 Cascade

Given that metastasis is one of the characteristics of CSCs, we investigated the effects of HK2 on sphere-forming abilities and changes to CSC-related genes in ovarian cancer cell lines following HK2 manipulation. HK2 silencing in ES-2 (Figure 2B) and OVCAR-3 cells (Appendix A) inhibited sphere-forming abilities, compared to control cells (Figure 4A), which was concomitant with a down-regulation of NANOG, octamer-binding transcription factor 4 (OCT4), Kruppel-like factor 4 (KLF4), SOX9, and c-kit (CD117) mRNA expression (Figure 4B). Flow cytometric and Western blot analyses further revealed a decline in CD177 in shHK2-OVCAR-3 cells (Figure 4C). Inversely, an increase in CD117 in HK2-overexpressing SKOV-3 cells was detected by flow cytometric analysis (Figure 4H). GEPIA further revealed a significant clinical correlation between NANOG, OCT4, KLF4, SOX9, and CD117 and HK2 expression (*p* < 0.05) in ovarian cancer clinical samples (Figure 4D). 2-DG (2 mM) also retarded sphere-forming abilities in OVCAR-3 and OVCA 433 cells, compared to control cells (Figure 4E), which was concomitant with a down-regulation of NANOG and SOX9 mRNA expression (Figure 4F). It has been recently established that FAK and MEK-1/ERK1/2 activation regulates CSC properties in breast [21,22,23] and liver [24,25] cancers. It is, therefore, worthwhile to investigate whether HK2-induces CSC properties via the FAK/ERK1/2 signaling cascade. Intriguingly, we found that FAK inhibitor 14 and U0126 blocked HK2-mediated sphere-forming abilities (Figure 4G), as well as decreased NANOG and SOX9 protein expression compared to control cells (Figure 4I).

### 2.6. Overexpression of HK2 in Ascites-Derived Sphere Forming Cells: HK2 Abrogation Impedes Lactate Production, Metastasis, and CSC Properties in Spheroids Formed from Ascites-Derived Tumor Cells

We further observed by qPCR a progressive increase in the levels of HK2 mRNA from normal ovarian epithelial cell lines to primary tumors to ascites (Figure 5A). In addition, a significantly higher expression of CSC-related genes (NANOG, OCT4, SOX2, KLF4) was found in spheroids formed from ascites-derived tumor cells, compared to monolayer cells by qPCR (Figure 5B). HK2 expression (Figure 5B) and lactate production (Figure 5C) were also augmented in spheroids. Knockdown of HK2 by siRNA in spheroids formed from ascites-derived tumor cells (Figure 5D) inhibited lactate production (Figure 5E), cell migration, invasion, sphere-formation abilities (Figure 5F), FAK/ERK1/2 activation (Figure 5G), and metastasis- and CSC-related gene expression (Figure 5H). 2-DG also retarded migration, invasion and sphere-forming abilities, compared to control cells (Figure 5F). These findings suggest that spheroids formed from ascites-derived tumor cells endure CSCs properties, and that this subpopulation of cells has enhanced glycolysis.

### 2.7. HK2 Heightens Tumor Growth and Dissemination in Nude Mice

To verify the effect of HK2 on in vivo tumor growth and dissemination, shHK2 ES-2 and control cells were inoculated s.c. or i.p. into nude mice. Significantly reduced tumor growth was detected in HK2 knockdown mice (Figure 6A). Fourteen days after i.p. inoculation, widespread abdominal dissemination, primarily in the mesentery, was observed in control mice. On the contrary, mice with HK2-silenced cells revealed only focal nodules in the mesentery (Figure 6B). The total i.p. tumor weight in the shHK2 ES-2 cell-injected mice (0.078 ± 0.031 g) was significantly lower than that of control mice (0.199 ± 0.066 g; *p* < 0.05).

### 2.8. CAF-Derived IL-6 Regulates HK2 in Ovarian Cancer Cells via IL6R

The tumor microenvironment is indispensable in ovarian cancer-metastatic niche formation, yet the underlying mechanisms remain unexplored [26]. CAF constitutes the main component of the tumor microenvironment [27]. To explore the possible upstream mechanisms leading to HK2 upregulation in ovarian cancer cells, we determined whether ovarian CAF-CM regulates HK2 expression in SKOV-3 cells. We found that CAF-CM attenuated HK2 expression (Figure 6C). CAF contributes to ovarian cancer progression and metastasis through secretion of chemokines and ECM [27]. Given that IL-6 is one of the major cytokines upregulated in CAFs [2,26], we found that CAF-CM-induced HK2 expression was abolished by neutralizing IL-6 and IL-6R antibodies (Figure 6C). Moreover, we revealed increased HK2 (Figure 6D) and lactate production in IL-6 treated SKOV-3 cells (Figure 6E), and a reversal of IL-6-induced HK2 was detected after treatment with neutralizing IL-6 and IL-6R antibodies (Figure 6D). We also investigated if IL-8, another major cytokine upregulated in CAFs [2,26], could regulate HK2 expression. Results revealed that neutralizing IL-8 and IL-8R antibodies could not block CAF-CM-induced HK2 expression (Figure 6C). Moreover, HK2 expression remained unchanged after IL-8 treatment in the presence or absence of neutralizing IL-8 and IL-8R antibodies (Figure 6D).

## 3. Discussion

Altered glucose metabolism is one of the hallmarks of cancer [8]. Due to the lower efficiency of ATP generation by aerobic glycolysis, cancer cells take up more glucose and produce more lactate than normal cells. Such increased glucose consumption distinguishes many human cancer cells from their normal counterparts, thus ^18^FDG-PET imaging, a measure of glucose consumption, is a common technology used to detect tumors in clinical use [9]. Studies on altered glucose metabolism in ovarian cancer are limited. It is known that more invasive ovarian cancer cells display higher glucose uptake and lactate production [28]. Moreover, lactate levels were shown to be increased in both primary and metastatic ovarian cancer when compared to normal ovarian tissue [15]. Our findings revealed higher HK2 expression in ovarian cancer, which was correlated with shorter overall and disease-free survival. These results suggest that HK2 is a significant prognostic marker in ovarian cancer. We further discovered that HK2 is a significant independent predictor of disease-free survival. Additionally, we show for the first time to the best of our knowledge that higher HK2 expression is present in metastatic foci compared to their matched primary tumors. Up-regulation of HK2 was also detected in ovarian ascitic fluid samples compared to primary tumor cells and HOSE cells. Moreover, knockdown of HK2 resulted in lower lactate production in ovarian cancer cells. All of these findings indicate that HK2 may be one of the glycolytic genes responsible for the metabolic switch from primary to metastatic tumors.

The above described HK2 expression patterns, combined with our in vitro and in vivo experiments on cell migration and invasion, highlight the contribution of HK2 to ovarian cancer metastasis. We also demonstrated a link between HK2 and FAK, MEK-1/ERK1/2, MMP9, uPA, and VEGF in the regulation of cell migration and invasion. Mechanistically, we have provided evidence that the activation of FAK/MEK-1/ERK1/2 signaling, which in turn induces MMP-9 expression, is involved in HK2-mediated regulation of cell migration and invasion. This pathway is linked to the formation and turnover of focal adhesions, which is vital for tumor cell invasion and metastasis [18], including in ovarian [29,30] and gastric [31] cancers. Moreover, uPA and VEGF expression have both been shown to be associated with tumor stage, metastasis and patient survival in ovarian cancer [32,33]. However, the mechanism by which HK2-mediates an up-regulation of uPA and VEGF needs to be elucidated in more detail. A humanized monoclonal antibody to VEGF-A, bevacizumab, is the only FDA approved anti-angiogenic agent for the treatment of ovarian cancer. However, a modest response to resistance limits its clinical efficacy [33]. The present link between HK2 and VEGF suggests that dual targeting of HK2 and VEGF in ovarian or other cancers may be a promising alternative therapeutic approach, which should be evaluated in future studies.

The progressive increase in HK2 levels from normal ovarian epithelial cells, to primary tumors, to ascites, as well as enhanced expression of HK2 and lactate production in ascites-derived sphere forming cells compared to monolayer cells, supports the notion that ascites bear CSC characteristics and harbor enhanced glycolysis [4]. Apart from the well-established effects on cellular metastasis, FAK has been shown to regulate the self-renewal and tumor-initiating capabilities of CSCs in many types of cancer, such as in breast cancer [21,22], Ductal Carcinoma In Situ [34], mesothelioma [35], liver cancer [24], and squamous cell carcinomas [36]. MEK/ERK activation has also been documented to be essential for mediating cancer stemness activities in breast [23] and liver [25] cancers. In line with this, the treatment of breast and liver cancer cell lines with FAK inhibitor 14 has been found to decrease NANOG expression [22,24]. In addition, ERK signaling was found to be involved in the up-regulation of SOX9 by fibroblast growth factors in chondrocytes [37]. In addition to the effects on cell migration and invasion, we believe our identification of the downstream pathways of HK2—Namely the HK2 to FAK to MEK-1/ERK1/2 pathway that controls NANOG and SOX9 expression and ovarian cancer CSC properties—Will be of significant importance to future studies. Besides NANOG and SOX9, we also revealed an up-regulation of OCT4, KLF4, and CD117 by HK2 in ovarian cancer, which further supports the concept that HK2 regulates CSCs.

The present link between HK2 and FAK/MEK-1/ERK1/2 activation, leading to MMP9/NANOG/SOX9 expression and increased ovarian cancer metastasis and CSC properties, suggests that targeting HK2/FAK/MEK-1/ERK1/2 signaling may be a promising therapeutic alternative, either as mono therapy or in combination with other treatments. 2-DG, a glucose analog, enters cells via GLUTs, thereby inhibiting glucose uptake. Once inside the cell, 2-DG is phosphorylated by HKs to form 2-DG-6-phosphate, which is not further metabolized. The latter accumulates and noncompetitively inhibits HKs and competitively inhibits PGI, thus halting glycolysis [38]. A phase I clinical trial of 2-DG alone or in combination with docetaxel in patients with advanced solid tumors has shown that 2-DG is a safe agent for clinical use [39]. Our data also showed a blockage of metastasis and sphere formation by 2-DG in vitro, suggesting the activity of HK2 is essential for its metastasis and stemness effects in ovarian cancer. Since 2-DG is considered as an effective glycolysis inhibitor that noncompetitively inhibits HKs, there is a need for the development of small molecule inhibitors that specifically inhibit HK2. Defactinib, a small-molecule and well-tolerated FAK inhibitor, has been evaluated in phase I/Ib study combined with paclitaxel in patients with relapsed ovarian cancer (NCT01778803), and in a phase II study in patients with the NSCLC KRAS mutation (NCT01951690). It is also currently being evaluated in a phase 1/1b study combined with avelumab, an anti-PD-L1 antibody, in patients with relapsed ovarian cancer (NCT02943317). In a phase 3 study, Trametinib, a selective MEK inhibitor, was shown to improve overall and progression-free survival in patients with metastatic melanoma, as compared to chemotherapy alone [40]. A phase 2 clinical trial of another MEK inhibitor, selumetinib, also displayed good tolerance and an active effect in patients with recurrent low-grade serous carcinoma of the ovary or peritoneum [41]. Taken together, targeting HK/FAK/MEK-1/ERK1/2 signaling either alone or in combination is possible, and can be evaluated for the future treatment of ovarian cancers.

Besides cell metastasis and CSC regulation, we also demonstrated enhanced cell proliferation and anchorage-independent growth of HK2 in ovarian cancer cells, in line with the well-established HK2 tumor-promoting effect observed in other cancers in vivo, such as glioblastoma [12], medulloblastoma [13], and breast cancer [14]. Lysophosphatidic acid, a blood-borne lipid mediator that is elevated in the ascites of ovarian cancer patients, has been shown to up-regulate HK2 and glycolysis, leading to enhanced proliferation of ovarian cancer cells [42]. A recent study also revealed that HK2 contributes to ovarian cancer cisplatin resistance by regulating cisplatin-induced, ERK-mediated autophagy [43].

Mounting evidence suggests that the tumor microenvironment plays an important role in ovarian cancer progression and metastasis [26]. CAFs transform from normal fibroblasts in the stroma after interacting with ovarian cancer cells, which constitutes more than half the tumor microenvironment [27]. CAFs plays important roles in cell growth, adhesion, invasion, and metastasis by secreting chemokines and ECM, facilitating dissemination [26]. In this study, we found that CAF-CM up-regulates HK2 in ovarian cancer. Moreover, we show that IL-6, but not IL-8, is one of the major cytokines in CAF-CM that contributes to HK2 up-regulation through binding of IL-6R in ovarian cancer cells, suggesting that HK2 expression can be regulated by the tumor microenvironment in ovarian cancer.

## 4. Materials and Methods

### 4.1. Clinical Samples

Seventy-five formalin-fixed, paraffin-embedded tumor samples from ovarian cancer patients, which included six benign ovarian cystadenomas, 45 primary ovarian carcinomas of different histological subtypes, and 24 matched metastatic foci, were retrieved from the Department of Pathology, Queen Mary Hospital, University of Hong Kong. All patients underwent surgery, and 41 further received chemotherapy, including platinum/paclitaxel. The follow-up period was 47 months (mean) and ranged from 5 to 84 months. Each sample was histologically examined to confirm the diagnosis. Fresh ovarian tumor specimens and ascitic fluid were obtained from patients with serous and clear cell ovarian cancer undergoing tumor-debulking surgery. For ovarian tumor specimens, solid tumor tissues were finely minced with scissors and submerged in PBS (Ca^2+^/Mg^2+^-free) with 1 mg/mL collagenase/dispase (Roche, Brighton, MA, USA). The mixture was stirred slowly for 80 min at 37 °C, and was then filtered through sterile cell strainers (40 μm, Corning, Tewksbury, MA, USA) and centrifuged at 100 G for 10 min. For ascitic fluid samples, cell pellets were obtained by centrifugation at 100 G for 10 min. Supernatants were discarded. Afterwards, tumor cells were purified with different solutions of NaCl to exclude erythrocytes. Isolated ovarian tumor cells and ascites-derived tumor cells were cultured in Medium 199 (Invitrogen, San Diego, CA, USA)/MCDB 105 (Sigma-Aldrich, St. Louis, MO, USA) and were supplemented with 10% fetal bovine serum and 100 U/mL penicillin and streptomycin (Invitrogen, San Diego, CA, USA). The use of samples was approved by the Institutional Ethical Review Board (UW 16-107). Informed written consent was obtained from all patients.

### 4.2. Cell Lines

Two immortalized ovarian epithelial cell lines (HOSE 6-3 and HOSE 11-12) and 12 ovarian cancer cell lines (SKOV-3, OVCAR-3, OVCA420, OVCA429, OVCA433, ES-2, TOV-21G, TOV112D, A2780S, and A2780CP) were cultured as previously described [44,45]. SKOV-3, OVCAR-3, ES-2, TOV-21G, and TOV112D were from the American Type Culture Collection (ATCC; Manassas, VA). HOSE 6-3, HOSE 11-12, OVCA420, OVCA429, and OVCA433 were given by Prof. S.W. Tsao (Department of Anatomy, University of Hong Kong, Hong Kong, China).

### 4.3. Spheroid Cultures

Cells were seeded as a single cell suspension in 6-well ultra-low attachment plates (Corning) at a density of 5000 cells/well in serum-free DMEM/F12 media (1:1) (Invitrogen), supplemented with basic fibroblast growth factor (10 ng/mL; Sigma-Aldrich), human epidermal growth factor (20 ng/mL; Sigma-Aldrich), and insulin (5 μg/mL; Sigma-Aldrich). Fresh media was added to each well every two days without removing the old media [46].

### 4.4. Transient and Stable Silencing of HK2 and 2-DG Treatment

For transient silencing, siRNAs specifically targeting HK2 and control siRNA (Invitrogen) were transfected into A2780CP, ES-2, and ascites-derived tumor cells using SilentFect (Bio-Rad Laboratories, Hercules, CA) for 48 h before cell counting and cell plating for subsequent assays. For stable silencing, A2780CP, ES-2, and OVCAR-3 cells were stably transfected with SureSilencing shRNA plasmids against human HK2 and a negative control plasmid (Qiagen, Valencia, CA, USA) using Lipofectamine 3000 (Invitrogen). Stable clones were selected using puromycin (1.5 µg /mL) [44,45]. For 2-DG treatment, OVCA 433 cells were plated 24 h before treatment with 2-deoxyglucose (2-DG; 2mM; Sigma-Aldrich) or vehicle (DMSO, Sigma-Aldrich) in complete medium (2 mM glucose). Cells were then harvested for immunoblot analyses [44,45].

### 4.5. Transient and Stable Overexpression of HK2 and Drug Treatments

SKOV-3 cells were transfected with pCMV6-DDK-HK2 and pCMV6-DDK (control vector; Origene, Rockville, MD, USA) using Lipofectamine 3000 (Invitrogen). Transiently transfected cells, transfected for 72 h, were counted and plated for subsequent assays. For stable overexpression, stable clones were selected using G418 (500 µg/mL). For drug treatment, HK2 stably-overexpressed cells cultured for 24 h were treated with the FAK inhibitor 14 (10 μM; Santa Cruz Biotechnology, Inc., Santa Cruz, CA, USA) and the MEK-1 inhibitor U0126 (10 μM; Sigma-Aldrich) for 48 h. Cells were then harvested for immunoblot analyses [44,45].

### 4.6. Immunohistochemistry

Immunohistochemistry was performed as described [44,45,47]. Briefly, formalin-fixed paraffin sections were stained with an antibody against HK2 (1:100; Appendix A) using the EnVision+ Dual Link System (K4061; Dako, Carpinteria, CA, USA). Antigen was retrieved by heating in 10 mM citrate sodium buffer (pH 6.0), heated in a microwave oven. Removal or substitution of the primary antibody with preimmune IgG serum was performed as a negative control. Both the intensity and percentage of stained epithelial cells were evaluated semi-quantitatively. Staining intensity was scored as 0 (negative), 1 (faint), 2 (moderate), or 3 (strong). The percentage of positive cells was assessed as 0 (<5%), 1 (5%–25%), 2 (26%–50%), 3 (51%–75%), or 4 (>75%). A composite “histoscore” was given by multiplying the staining intensity (0–3) by the percentage of positive cells (0–4). High and low expression of HK2 was expressed by mean histoscore cut-offs.

### 4.7. Real-Time PCR (qPCR)

Real-time PCR was performed using cDNA synthesized from the total RNA extracted from ovarian tumors, ascites-derived tumor cells, and cancer cell lines, using Power SYBR Green PCR Master Mix (Invitrogen, San Diego, CA, USA) with the ABI Prism 7700 sequence detection system (Invitrogen) [44,45]. Primer sequences are listed in Appendix A. 

### 4.8. Clinical Correlations

Correlations between target genes and HK2 expression in ovarian cancer in the TGCA database cohort was determined using the GEPIA (http://gepia.cancer-pku.cn/) tool.

### 4.9. Immunoblotting

Protein lysate (20 μg) was resolved by SDS-PAGE, transferred to polyvinylidene difluoride membranes, and hybridized with the corresponding primary antibodies, as listed in Appendix A, followed by the corresponding secondary antibodies (Santa Cruz) [44,45,47].

### 4.10. Flow Cytometry

Single-cell suspensions obtained by filtering through a 40 μm cell-strainer were labeled with a PE-conjugated CD117 antibody (BD Biosciences, San Jose, CA, USA). The percentage of positive cells was evaluated by flow cytometry analysis in a BD FACSAria flow cytometer (BD Biosciences).

### 4.11. Lactate Assay

Lactate levels were measured in the media from cultured cells using the Lactate Colorimetric Assay Kit II (BioVision, Milpitas, CA, USA), and were normalized to cell number.

### 4.12. Wound Healing Assay and In Vitro Migration and Invasion Assays

For the wound healing assay, a wound was made in cells plated in 6-well plates for 24 h using a sterile pipette tip, followed by replacement with fresh culture media. The same position of the wound was photographed at time 0 and 7 h (for ES-2), or 23 h (for A2780CP) [44]. In vitro migration and invasion assays were performed as previously described [44,45,46]. Cells at a density of 1.25 × 10^5^ were plated on the upper compartment of a Transwell chamber, and were allowed to migrate through a membrane (8 µm pore size)/invaded through a Matrigel–coated membrane. After 12 to 48 h, cells on the upper side of the membrane were removed. The migrated/invaded cells were fixed, stained, and counted. For drug treatment, cells were plated on the upper side of a Transwell chamber for 6 h before treatment with 2-DG (0, 0.5 and 2 mM), FAK inhibitor 14 (10 µM), U0126 (10 µM), or vehicle. For treatment using the anti-MMP9 neutralization antibody, cells were pre-treated with the neutralization antibody or the mouse IgG for overnight before cell counting and cell plating.

### 4.13. XTT Assay, Cell Counting, and Soft Agar Assay

For the XTT assay, 2000 cells/well were seeded in 96-well plates. Five days after incubation, cell proliferation was measured using the Cell Proliferation Kit II (Roche) according to the manufacturer’s instructions in an Infinite^®^ 200 microplate reader at 492 nm (Tecan Group Ltd., Männedorf, Switzerland) [46]. For cell counting, 3 × 10^4^ cells were seeded in 12- or 6-well plates or T150 culture flasks, and maintained in growth media. Cell numbers were counted at days 1 (12-well culture plates), 4 (6-well culture plates), 7, and 11 (T150 culture flasks), using a Luna™ automated cell counter (Logos Biosystems, Annandale, VA, USA) [44]. For the soft agar assay, 2 × 10^4^ cells were suspended in 2 mL 0.4% agar and seeded on 1% agar in 6-well plates. After 4 weeks, cells were counted.

### 4.14. Sphere-Formation Assay

Cells were seeded and cultured as spheroid cultures, as above. After 9–13 days, spheres were counted and imaged under an inverted microscope. Spheres that <50 μm and individual or aggregated cells were not counted as spheres. For drug treatment, plated cells were treated with 2-DG (0 and 2 mM), FAK inhibitor 14 (10 µM), U0126 (10 µM), or vehicle. 

### 4.15. In Vivo Analyses

2 × 10^6^ ES-2 cells with stably knocked-down HK2 were injected subcutaneously (s.c.; five mice/group) or intraperitoneally (i.p.; seven mice/group) into BALB/c female nude mice [44]. Perpendicular tumor diameters were measured on days 7, 11, and 14, and tumor volumes were calculated. Fourteen days post cell injection, mice were sacrificed, and tumor dissemination was recorded. Experiments were performed following the Animals (Control of Experiments) Ordinance (Hong Kong) and the Institute’s guidance on animal experiments.

### 4.16. Treatment with Neutralizing Antibodies in Combination with CAFs-CM, IL-6 or IL-8

Serum starved SKOV-3 cells were treated for 48 h with neutralizing antibodies against IL-6, IL-6R, IL-8, IL-8R or mouse IgG (Appendix A), in combination with CAFs-CM (Vitro Biopharma, CO, USA), recombinant human IL-6 (50 ng/mL R&D Systems, Minneapolis, MN, USA) or recombinant human IL-8 (50 ng/mL; R&D Systems) for 48 h [44].

### 4.17. Statistical Analyses

Statistical analyses were performed using SPSS 20 for Windows (SPSS Inc., Chicago, IL, USA). The Mann-Whitney test was used for comparing data between two groups while the Kruskal–Wallis rank test was performed for multiple comparisons. Kaplan–Meier and log-rank tests were used for survival analyses. Multivariate survival analyses were performed using a Cox regression analysis. *p* values <0.05 were considered statistically significant.

## 5. Conclusions

In conclusion, we describe in this study the overexpression of HK2 in ovarian cancer. Notably, high HK2 expression was associated with cancer metastasis and poor patient clinical outcomes. CAFs secrete IL-6 in the tumor microenvironment, which contributes to an up-regulation of HK2 via the IL-6R (Figure 6F). We also revealed that HK2 controls a metabolic switch, which promotes cell migration, invasion, CSC properties, proliferation, and anchorage-independent growth in ovarian cancer cells. We demonstrated that the mechanisms regulating cell migration/invasion and CSC properties involve FAK/MEK-1/ERK1/2/MMP9/NANOG/SOX9 signaling (Figure 6F). Improved understanding of the involvement of HK2 in metastasis and in CSC properties will facilitate its effective application as a therapeutic molecular target, either alone or in combination with other treatments.

## Figures and Tables

**Figure 1 cancers-11-00813-f001:**
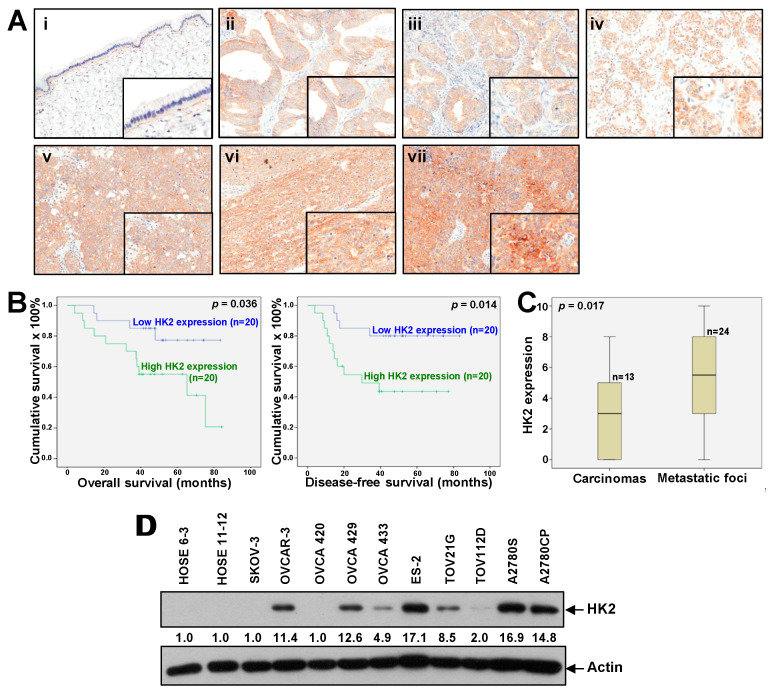
Up-regulated HK2 in ovarian cancer is linked to tumor metastasis and poor survival. (**A**) Immunohistochemical staining of HK2 in mucinous benign cystadenoma (**i**); mucinous (**ii**), endometrioid (**iii**), and clear cell (**iv**) carcinomas; primary serous carcinomas (**v**); and matched metastatic foci (**vi**) and (**vii**). Magnification: 20×. The insets highlight regions with higher magnification. (**B**) Kaplan–Meier overall (left panel) and disease-free (right panel) survival curves for ovarian cancer patients with low and high HK2 expression levels (cut-off at mean). (**C**) HK2 immuno-scoring in primary carcinomas and corresponding metastatic foci. (**D**) HK2 protein expression in normal ovarian epithelial cell lines (HOSE) and ovarian cancer cell lines as assessed by immunoblot analysis. Densitometric analysis is shown normalized to Actin.

**Figure 2 cancers-11-00813-f002:**
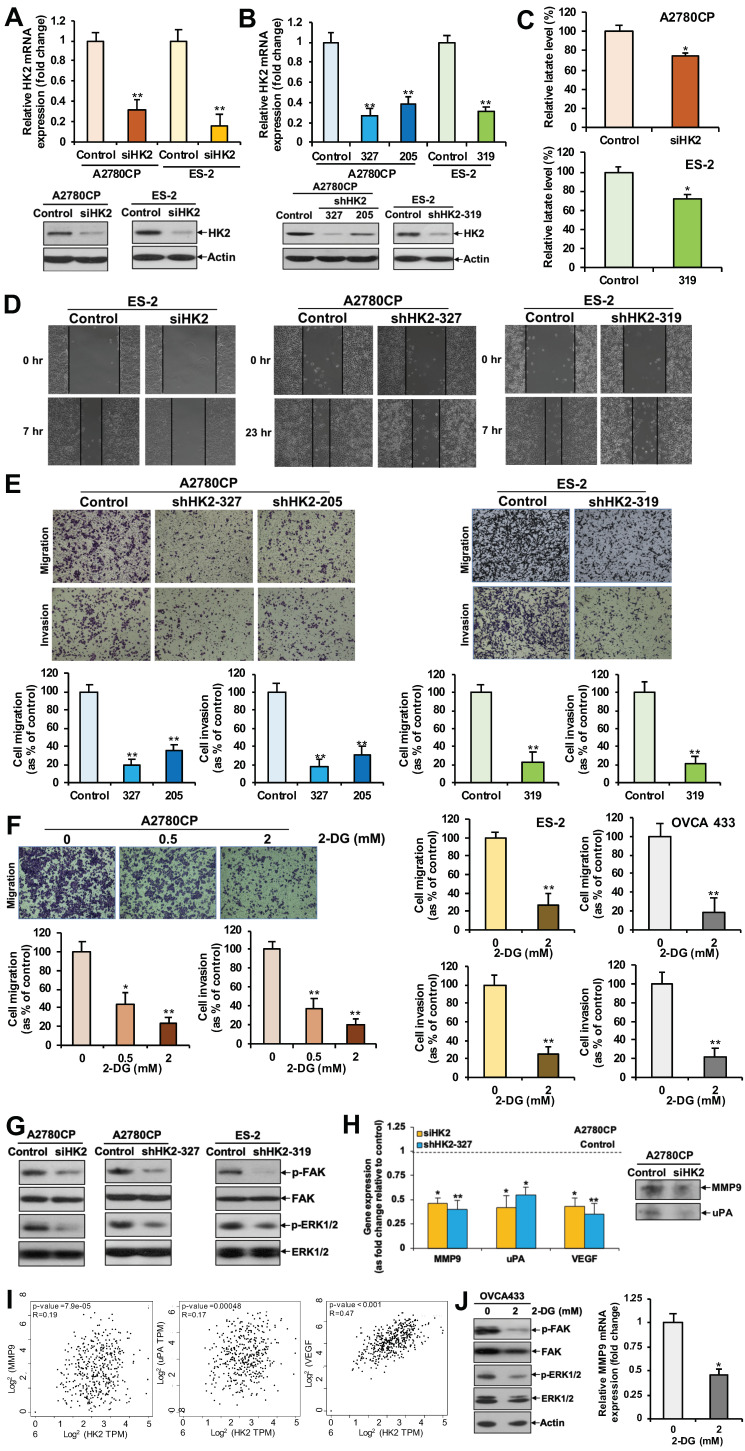
HK2 depletion hinders lactate production, impedes ovarian cancer cell migration and invasion, and reduces FAK and ERK1/2 activation, as well as MMP9, uPA and VEGF expression. (**A**) Transient knockdown of HK2 (via siHK2) mRNA and protein expression in A2780CP and ES-2 cells, as determined by qPCR (upper panel) and immunoblot analysis (lower panel), respectively. (**B**) Stable knockdown of HK2 (shHK2) mRNA and protein expression in A2780CP and ES-2 cells, as determined by qPCR (upper panel) and immunoblot analysis (lower panel), respectively. (**C**) Fold change in lactate levels in siHK2 (A2780CP), shHK2 (ES-2), and control cells, as assessed using a lactate colorimetric assay. *n* = 3; *, *p* < 0.05. (**D**) Wound healing assay in control conditions and after transient/stable knockdown of HK2 in A2780CP and ES-2 cells. (**E**) Migration or invasion of A2780CP and ES-2 cells with stable knockdown of HK2 (shHK2), presented as a percentage of controls; *n* = 3; **, *p* < 0.005. Representative images of migrating or invading A2780CP and ES-2 cells (upper panel). (**F**) Migration or invasion of 2-DG-treated and control A2780CP, ES-2 and OVCA 433 cells, presented as a percentage of controls; *n* = 3; *, *p* < 0.05; **, *p* < 0.005. Representative images of migrating A2780CP cells (left upper panel). (**G**) Immunoblot analyses of FAK and ERK1/2 activation in HK2-transiently/stably silenced A2780CP and ES-2 cells. (**H**) (left panel) mRNA expression of MMP9, uPA, and VEGF, calculated as fold change in HK2-transiently/stably silenced and control A2780CP cells using qPCR; *n* = 3; *, *p* < 0.05; **, *p* < 0.005. (**H**) (right panel) Immunoblot analyses of MMP9 and uPA expression in conditioned media obtained from control and siHK2 A2780CP cells. (**I**) Correlation between MMP9, uPA, and VEGF, and HK2 in ovarian cancer patients in TGCA database cohorts using the GEPIA tool. (**J**) Immunoblot analyses of FAK and ERK1/2 activation (left panel), and mRNA expression of MMP9 calculated as fold change using qPCR (right panel) in 2-DG treated OVCA 433 cells; *n* = 3; *, *p* < 0.05.

**Figure 3 cancers-11-00813-f003:**
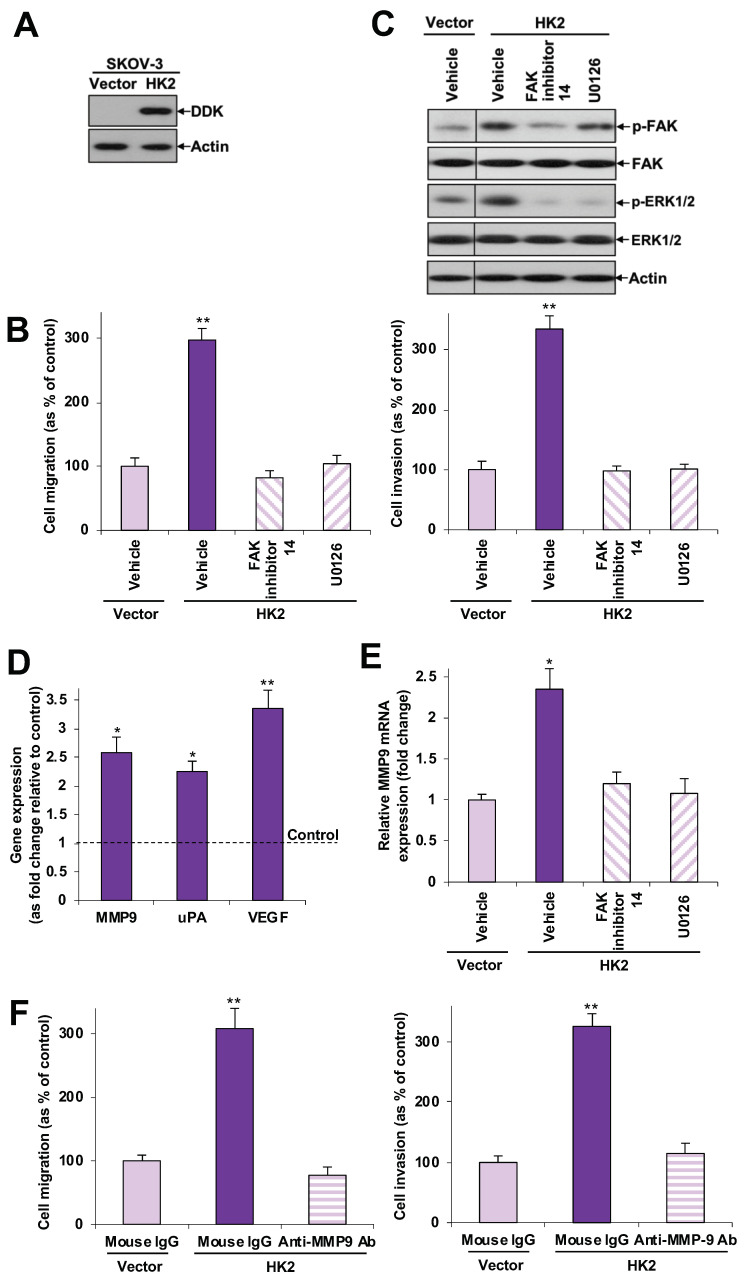
HK2-mediated cell migration and invasion involves the FAK/MEK-1/ERK1/2/MMP-9 pathway. (**A**) Immunoblot analysis of exogenous DDK-tagged HK2 expression in SKOV-3 cells stably transfected with control vector pCMV6-DDK or DDK-tagged HK2. (**B**) In vitro migration and invasion assays in SKOV-3 cells expressing HK2 in the presence or absence of FAK inhibitor 14 or U0126. Cell migration and invasion are presented as a percentage of controls; *n* = 3; **, *p* < 0.005. (**C**) Immunoblot analysis of FAK and ERK1/2 activation in SKOV-3 cells expressing HK2 in the presence or absence of FAK inhibitor 14 or U0126. (**D**) mRNA expression of MMP9, uPA and VEGF, calculated as fold change in HK2-overexpressing SKOV-3 cells, using qPCR; *n* = 3; *, *p* < 0.05; **, *p* < 0.005. (**E**) mRNA expression of MMP9 in SKOV-3 cells expressing HK2 in the presence or absence of FAK inhibitor 14 or U0126, using qPCR; *n* = 3; *, *p* < 0.05. (**F**) In vitro migration and invasion assays in SKOV-3 cells expressing HK2 in the presence or absence of pre-treatment with an anti-MMP9 neutralization antibody. Cell migration and invasion are presented as a percentage of controls; *n* = 3; **, *p* < 0.005.

**Figure 4 cancers-11-00813-f004:**
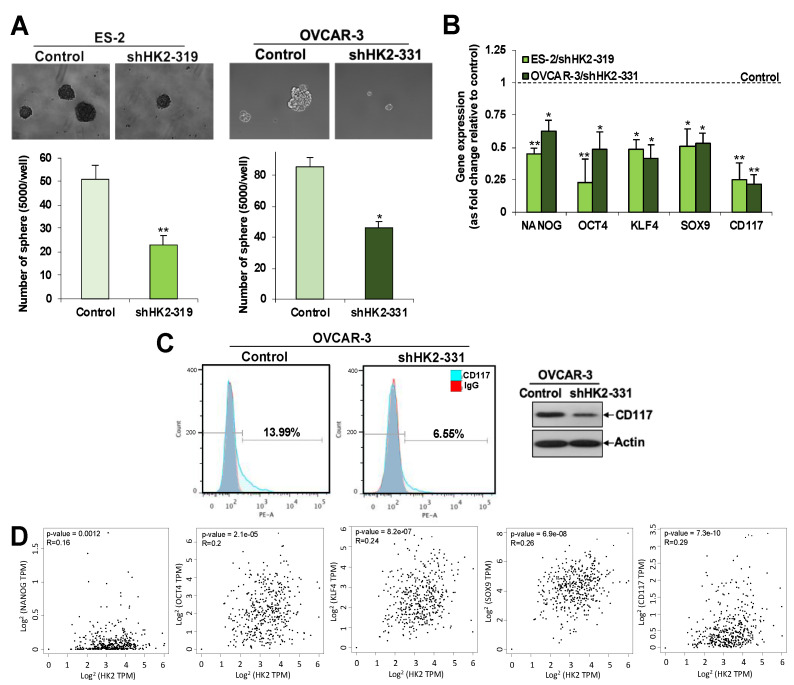
HK2 enhances CSC properties via the FAK/MEK-1/ERK/NANOG/SOX9 pathway. (**A**) Sphere-forming assay in control and HK2-silenced conditions in ES-2 and OVCAR-3 cells. Sphere formation is presented as the number of spheres formed. *n* = 3; *, *p* < 0.05; **, *p* < 0.005. Representative images of sphere forming cells formed (upper panel). (**B**) mRNA expression of NANOG, OCT4. KLF4, SOX9, and CD117, calculated as fold change in HK2-overexpressing and control SKOV-3 cells, using qPCR; *n* = 3; *, *p* < 0.05; **, *p* < 0.005. (**C**) Flow cytometry (left) and Western blot (right) analyses of the cell surface marker CD117 in control and HK2-silenced OVCAR-3 cells. (**D**) Correlation between NANOG, OCT4. KLF4, SOX9 and CD117, and HK2 in ovarian cancer patients in TGCA database cohorts using the GEPIA tool. (**E**) Sphere-forming assay in OVCAR-3 and OVCA 433 cells treated with 2 mM 2-DG or vehicle. *n* = 3; *, *p* < 0.05. (**F**) mRNA expression of NANOG and SOX9, calculated as fold change in 2-DG treated OVCA 433 cells, using qPCR; *n* = 3; *, *p* < 0.05. (**G**) Sphere-forming assay in SKOV-3 cells expressing HK2 in the presence or absence of FAK inhibitor 14 or U0126. Sphere formation is presented as the number of spheres formed. *n* = 3; *, *p* < 0.05. (**H**) Flow cytometry analysis of the cell surface marker CD117 in control and HK2-overexpressing SKOV-3 cells. (**I**) Immunoblot analysis of NANOG and SOX9 in SKOV-3 cells expressing HK2 in the presence or absence of FAK inhibitor 14 or U0126.

**Figure 5 cancers-11-00813-f005:**
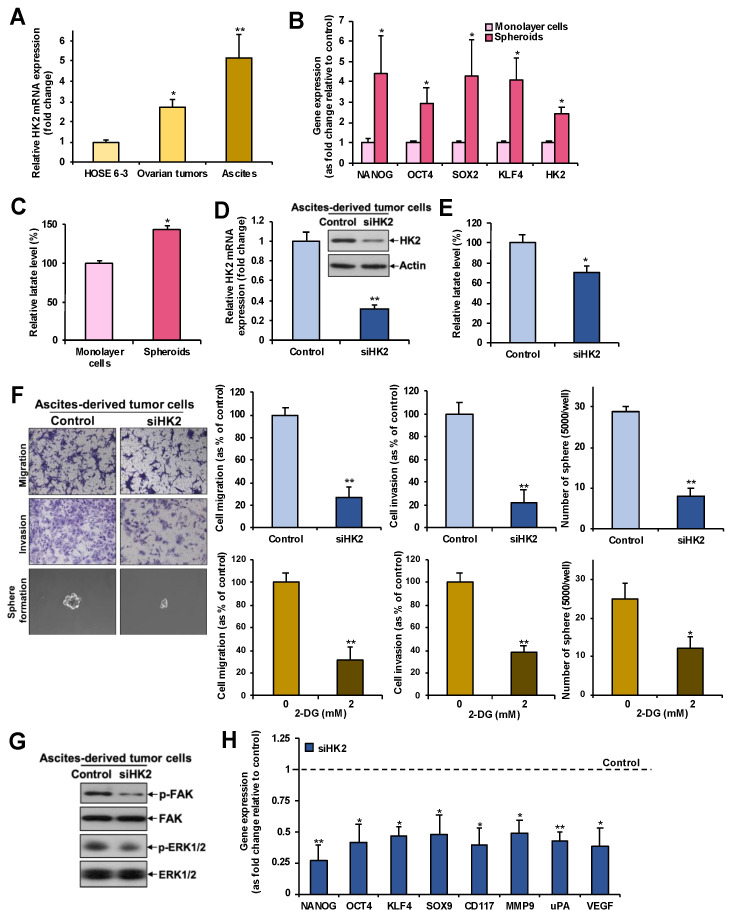
Increased expression of HK2 and lactate production in ascites-derived spheroids compared to monolayer cells: Silencing of HK2 in spheroids formed from ascites-derived tumor cells results in a reduction in lactate production, metastasis and CSC properties. (**A**) mRNA expression of HK2 in a normal ovarian epithelial cell line, primary tumors and paired ascites (*n* = 3), as determined by qPCR; *, *p* < 0.05 and **, *p* < 0.005, compared to HOSE 6-3 cells. (**B**) mRNA expression of NANOG, OCT4, SOX2, KLF4 and HK2 in spheroids compared to monolayer cells formed from ascites-derived tumor cells, as determined by qPCR. *n* = 5; *, *p* < 0.05. (**C**) Fold change of lactate level in spheroids compared to monolayer cells formed from ascites-derived tumor cells as assessed using a lactate colorimetric assay. *n* = 3; *, *p* < 0.05. (**D**) Transient knockdown of HK2 (via siHK2) mRNA and protein expression in spheroids formed from ascites-derived tumor cells, as determined by qPCR (upper panel) and immunoblot analysis (lower panel), respectively. (**E**) Fold change in lactate levels in transiently silenced HK2 cells compared to controls, in spheroids formed from ascites-derived tumor cells, as assessed using a lactate colorimetric assay. *n* = 3; *, *p* < 0.05. (**F**) Migration, invasion or sphere-forming abilities in spheroids formed from ascites-derived tumor cells with transient knockdown of HK2 (upper right) or treated with 2 mM 2-DG or vehicle (lower right) presented as a percentage of controls; *n* = 3; **, *p* < 0.005. Representative images of migrating, invading or sphere-forming cells with transient knockdown of HK2 (left). (**G**) Immunoblot analysis of FAK and ERK1/2 activation in spheroids formed from ascites-derived tumor cells with transient knockdown of HK2. (**H**) mRNA expression of NANOG, OCT4, KLF4, SOX9, CD117, MMP-9, uPA, and VEGF, in spheroids formed from ascites-derived tumor cells with transient knockdown of HK2, as determined by qPCR. *n* = 5; *, *p* < 0.05; **, *p* < 0.005.

**Figure 6 cancers-11-00813-f006:**
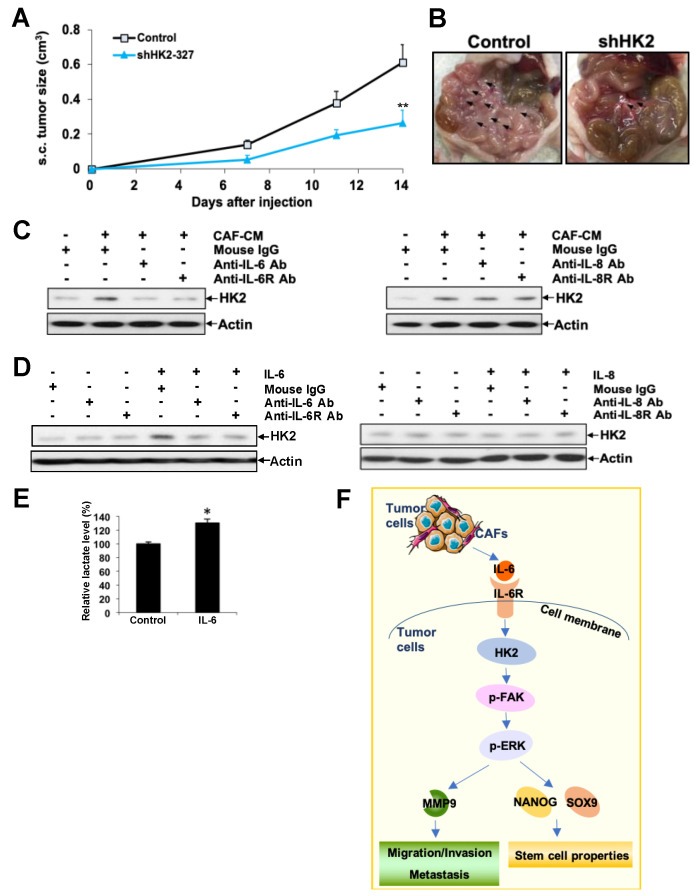
HK2 depletion hinders tumor growth and dissemination in nude mice: Conditioned media derived from ovarian cancer-associated fibroblasts (CAF-CM) induces HK2 expression in ovarian cancer cells through IL-6/IL-6R. (**A**) Growth rates of s.c. tumors formed in mice inoculated with shHK2 ES-2 or control cells (2 × 10^6^). (**B**) Representative views of the abdominal cavity of mice inoculated i.p. with shHK2 ES-2 or control cells. Arrows indicate tumors. (**C**) Immunoblot analysis of HK2 expression in serum-starved SKOV-3 cells treated with CAF-CM in the presence or absence of neutralizing antibodies to IL-6 and IL-6R (left), IL-8 and IL-8R (right) or corresponding control IgG, for 48 h. (**D**) Immunoblot analysis of HK2 expression in serum-starved SKOV-3 cells treated with IL-6 in the presence or absence of neutralizing antibodies to IL-6 and IL-6R (left), IL-8 in the presence or absence of neutralizing antibodies to IL-8 and IL-8R (right), or corresponding control IgG, for 48 h. (**E**) Fold change of lactate level in IL-6 treated SKOV-3 cells compared to control cells as assessed using a lactate colorimetric assay. *n* = 3; *, *p* < 0.05. (**F**) Schematic illustration showing that CAFs in the tumor microenvironment induces HK2 through IL-6/IL-6R. HK2 in turn, enhances cell migration, invasion, and CSC properties in ovarian cancer cells. The underlying mechanisms involve the activation of FAK/MEK-1/ERK1/2/MMP9/NANOG/SOX9 signaling pathways.

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
