# Peer review of "Hexokinase 2 Regulates Ovarian Cancer Cell Migration, Invasion and Stemness via FAK/ERK1/2/MMP9/NANOG/SOX9 Signaling Cascades"

_cancers, 2019, doi:10.3390/cancers11060813_

Round 1
Reviewer 1 Report
Overall the authors have responded to most comments. There are not further major comments, however one comment on the data in fugre 4c.
The data in figure 4C is not very convincing. Not sure how significant this small change is. It would have been great if authors could have done IF for CD117, to confirm their results. Not convinced with the current results in 4C and its interpretation. The small change in CD117 could differ so much just based on how we do gating in FACS.
Author Response
Response to Reviewer 1
Reviewer 1:
|
Comment 1:Overall the authors have responded most comments. There are not further major comments, however one comment on the data in figure 4c. The data in figure 4C is not very convincing. Not sure how significant this small change is. It would have been great if authors could have done IF for CD117, to confirm their results. Not convinced with the current results in 4C and its interpretation. The small change in CD117 could differ so much just based on how we do gating in FACS.
|
Response to Comment 1: We appreciate the comment of the reviewer. Besides flow cytometric analysis, Western blot analysis was performed. Result revealed a decline in CD177 in shHK2-OVCAR-3 cells (Fig. 4C). Such information is added in Result section, lines 219-220, Page 8.
|

Reviewer 2 Report
The authors performed several new experiments resulting in manuscript's improvement.
However, I still feel that the IL-6 results presented in Figure 6 are too preliminary. The paper form Nieman et al (Nat Medicine 2011; Ref. 2) that the authors cite suggests that inhibition of IL-8R reduces homing of ovarian cancer cells at a greater extent than inhibition of IL-6R. In Figure 6 the authors should at least perform inhibition assays with a-IL-8 and a-IL-8R antibodies to prove that IL-6 is the main player in the regulation of HKII.
Author Response
Response to Reviewer 2
Reviewer 2: |
Comment 1: The authors performed several new experiments resulting in manuscript’s improvement. However, I still feel that the IL-6 results presented in Figure 6 are too preliminary. The paper from Nieman et al (Nat Medicine 2011;Ref 2) that the authors cite suggests that inhibition of IL-8R reduces homing of ovarian cancer cells at a greater extent than inhibition of IL-6R. In Figure 6 the authors should at least perform inhibition assays with a-IL-8 and a-IL-8R antibodies to prove that IL-6 is the main player in the regulation of HKII.
|
Response to Comment 1: We appreciate the comment of the reviewer. We have performed inhibition assays with a-IL-8 and a-IL-8R antibodies. Result revealed that neutralizing IL-8 and IL-8Rantibodies could not block CAF-CM-induced HK2 expression (Fig. 6C). Moreover, HK2 expression remained unchanged after IL-8 treatment in the presence or absence of neutralizing IL-8 and IL-8Rantibodies (Fig. 6D). Such information is added in Result section, lines 324-328, Page 14. In this study, we found that CAF-CM up-regulates HK2 in ovarian cancer. Moreover, we show that IL-6, but not IL-8, is one of the major cytokines in CAF-CM that contributes to HK2 up-regulation through binding of IL-6R in ovarian cancer cells, suggesting that HK2 expression can beregulated by the tumor microenvironment in ovarian cancer. Such information is added in Discussion section, lines 413-417, Page 16.
|

Round 2
Reviewer 2 Report
The authors have performed the requested experiments, providing evidence for an involvement of IL-6 but not IL-8 regulates HK-II in ovarian cancer cells.
This manuscript is a resubmission of an earlier submission. The following is a list of the peer review reports and author responses from that submission.
Round 1
Reviewer 1 Report
In this manuscript authors have attempted to decipher the role of HK2 in ovarian cancer tumorigenic properties such as migration and invasion. Authors suggested the molecular HK2 driven molecular mechanism of these observed phenotypes. The quality of data in this manuscript is high and the overall presentation looks great. However, there are several areas in which efforts need to be made to enhance the quality of this manuscript.
A recent publication (https://www.ncbi.nlm.nih.gov/pubmed/29247711) has showed the role of HK2 in ovarian cancer drug resistance. Some parts of the assays in this manuscript are similar to the published article. The published study also showed the regulation of ERK1/2 by HK2.
Other Comments:
1. Did the authors look at the normal fallopian tube epithelium for HK2 staining? If not then why?
2. It's not clear how and why did the authors coose to do FAK, ERK1/2, MMP9, uPA etc? What was the rational to choose these, how did the authors arrive at the decision to look into these proteins? Did the authors followed already published studies? If they so then that need to be stated clearly.
3. The molecular biology in the manuscript is high quality however a big concern in the study is the choice of cell lines. A2780 and SKOV3 cells need to be used carefully because they are not good model cell lines. See Domcke et al. 2013, Nat. comm. (PMID 23839242). The histological origin of these lines has not been well documented. What was authors' reasoning to select these cell lines?
4. Visually shift in the peak doesn't look significant in figure 4C. It seems like over interpretation of the data.
5. How did the authors are define CSC properties? Did they look at any particular surface markers that are defined for CSCs? Did the authors look at drug resistance?
6. Although the data looks good and convincing, however, the connections are missing throughout the results sections.. it is not clear at all that how did the authors (rationale) choose to do what they did in the following results section.. such as, how did they think of CAF-CM then suddenly IL6. What were the reasons to follow on such things. There is a disconnection.
7. Control/comparison: It would have been valuable if authors could have done the same experiments with a FTSEC (normal HGSOC eg. FT190, FT282, FT33 etc) line and/or a HOSE line by either overexpressing or knockdown of HK2 depending on its status in these cell lines. Can the authors perform a few of these assays in any of the FTSEC or HOSE line? That would certainly be valuable for the manuscript.
Reviewer 2 Report
In this manuscript the authors demonstrate that hexokinase II is upregulated in ovarian cancer and metastatic foci and that its expression inversely correlates with ovarian cancer patients survival. They show that knockdown of HK-II impairs cancer cell invasion and migration, tumor formation and metastasis. Finally, they demonstrate that HK-II promotes cancer cell migration and invasion through a FAK/ERK1/2-dependent mechanism.
Major points
The authors used the HK-II inhibitor 2-DG only in in vitro migration/invasion assays. They should include experiments with 2-DG in Figure 2G-I, Figure 4A-C, Figure 5 and (if possible) in Figure 6A-B. This would give more strength to the clinical relevance of the study and will demonstrate whether the activity of HK-II or its presence per se' (structural factor) is essential for its pro-metastatic activity in ovarian cancer.
I feel that the data on IL-6 in figure 6 are too preliminar. I suggest removing them from this manuscript.